# Risk factors for pulmonary infection in elderly patients with type 2 diabetes: A protocol for systematic review and meta-analysis

**Zhaoyang Wei, Wenhao Su, Hairong Jia, Luo Yang, Jiaqi Zhang, Yanru Wang** ⃝ *

Department of Nursing, Zhejiang Chinese Medical University, Hangzhou, Zhejiang, China

* Wangyanru001@outlook.com

## Abstract

### Background

Lung infection is a prevalent chronic consequence of diabetes. Abnormal blood sugar levels, vascular endothelial damage, and alterations in capillary permeability predispose diabetes patients to lung infections. Currently, there is no comprehensive review addressing the risk factors for lung infection in diabetes. Consequently, our objective is to conduct a systematic review of the existing risk factors for lung infection in diabetes and offer recommendations for the targeted enhancement of treatment strategies.

### Methods and analysis

We will search five English literature databases (PubMed, Embase, Web of Science, CINAHL, and Cochrane Library) and 4 Chinese databases (CNKI, WanFang, SinoMed and VIP) since the founding of the database until December 01, 2024. We will perform a systematic examination and meta-analysis of cohort, case-control and cross-sectional studies to identify all population-based risk factors for diabetes patients with pulmonary infection. Two researchers will independently assess the publication, extract data, and evaluate the quality and potential biases present in the study. We will utilize RevMan 5.4 software and STATA 16.0 for data analysis. The included studies will be assessed using the Newcastle Ottawa Quality Assessment Instrument (NOS) and Agency for Healthcare Research and Quality (AHRQ). If the heterogeneity of the included studies is excessively high, we will perform subgroup and sensitivity analysis to identify probable sources of heterogeneity. The assessment of publication bias will be conducted using a funnel plot. Furthermore, we will employ the Grading of Recommendations Assessment, Development, and Evaluation (GRADE) approach to assess the quality of evidence for each exposure and outcome of interest.

**Data availability statement:** No datasets were generated or analysed during the current study. All relevant data from this study will be made available upon study completion.

**Funding:** The author(s) received no specific funding for this work.

**Competing interests:** The authors have declared that no competing interests exist.

## Discussion

This article introduces a research protocol to explore the influencing factors of pulmonary infection in diabetes. The results of this study will summarize the evidence of influencing factors of pulmonary infection in diabetes at present. We hope to provide reliable advice for clinicians to make decisions, so as to support the implementation of effective prevention strategies for diabetes pulmonary infection.

## Trial Registration

PROSPERO CRD42024606429

## Introduction

Diabetes is a metabolic disease characterized by abnormal blood sugar [1]. There will be 529 million diabetes patients in the world in 2021. It is estimated that the global prevalence of diabetes will reach 12.2% in 2045 [2]. Studies have shown that patients with diabetes are more prone to various cardiovascular diseases, such as macrovascular diseases (including coronary heart disease, stroke and peripheral vascular diseases), microvascular diseases (including end-stage renal disease, retinopathy and neuropathy) [3], cognitive impairment [4], urinary tract infection [5], etc.

Compared with non-diabetes patients, diabetes patients have poorer pulmonary function and are more likely to deteriorate [6,7]. The reason may be that sustained hyperglycemia leads to changes in capillary permeability and pulmonary microvascular disease, making patients more susceptible to diseases such as pneumonia and lung infections [8,9]. Among them, elderly diabetes patients are more likely to have infectious diseases, such as asymptomatic bacteriuria, urinary tract infection, lower limb infection, etc. [10–13]. Patients with type 2 diabetes complicated with pulmonary infection have difficulty in clinical control, which may lead to cardiopulmonary failure, and severe infection may lead to death of patients [14]. Some studies have shown that diabetes patients with lung infection have more serious clinical symptoms, longer treatment time, higher complication rate and mortality [15,16]. Therefore, determining the risk factors of pulmonary infection in diabetes patients is of great significance to reduce the incidence of pulmonary infection, improve the quality of life of patients and improve the prognosis of patients.

Therefore, this study aims to find out the risk factors of lung infection in diabetes patients through meta-analysis. To provide suggestion for clinical prevention of pulmonary infection in patients with diabetes.

## Methods

### Ethics and dissemination

This study doesn't require patient informed consent or approval from the ethics committee. The findings of the systematic review and meta-analysis will be shared in peer-reviewed journals.

## Study registration

This research protocol has been registered with PROSPERO (CRD42024606429), and we will follow the PRISMA-P guidelines [17] (Fig 1) according to the PRISMA statement [18].

## Eligibility criteria

**Participants.** This study was based on the Chinese Consensus for the Diagnosis and Treatment of Diabetes Complicated with Pneumonia. Patients with confirmed diabetes-related pulmonary infections through clinical diagnosis or self-reporting were included. To standardize the management of heterogeneity in diagnostic criteria, we required that the diagnostic basis of the patients must include: 1. Fasting blood glucose ≥ 7.0 mmol/L or random blood glucose ≥ 11.1 mmol/L and confirmed pulmonary infection by CT; 2. The clinical case needs to include CURB-65 score, inflammatory markers, and qualified pathogen evidence.

**Exposure.** The primary outcome measure will be participants basic characteristics that may serve as risk factors/ predictors of deterioration. These may include but are not limited to demographic characteristics (such as age, gender,

**PRISMA 2020 flow diagram for new systematic reviews which included searches of databases and registers only**

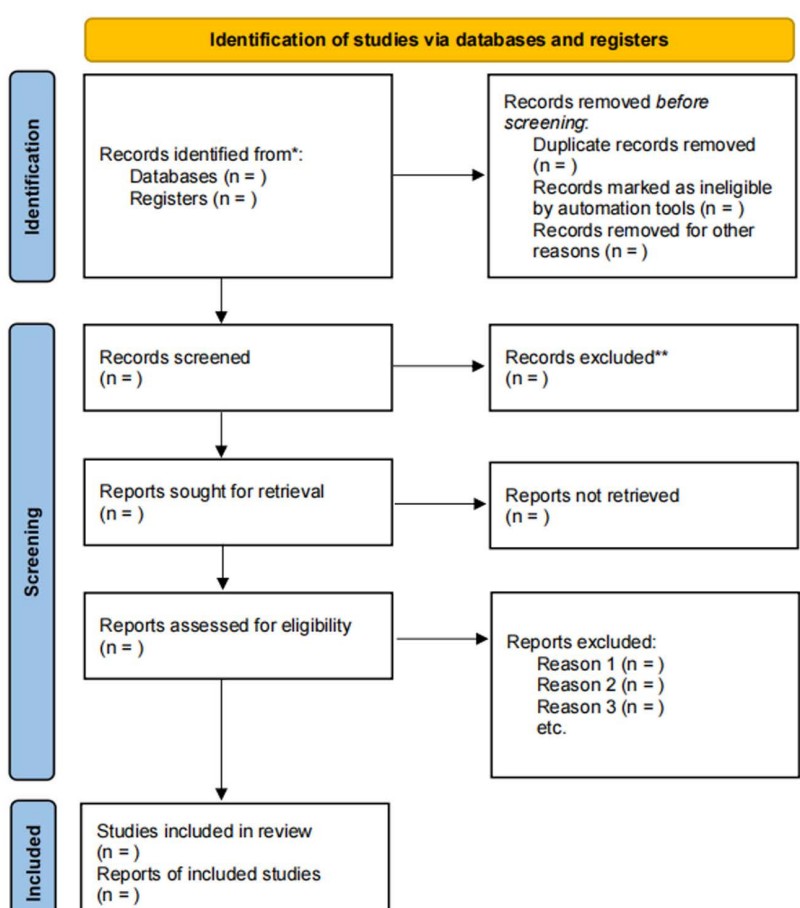

**Fig 1. Flowchart of studies included in the systematic review.**

race/ethnicity), characteristics related to the patient's pulmonary infection (such as disease duration, age, respiratory function, lung capacity), and other health-related characteristics (such as smoking, body mass index (BMI), blood glucose levels, comorbidities, and concomitant medications).

**Types of studies.** Only case-control studies, cohort studies and cross-sectional study will be considered.

## Exclusion criteria

We will exclude case control studies, cohort studies and cross-sectional studies of participants whose main health problem is not diabetes pulmonary infection, as well as studies that do not retain complete research data.

If the research meets the following criteria, it will be excluded:

(1) Repeated publications, conference articles, meta-analyses, reviews, protocols, animal studies, and letters;

(2) Unable to obtain the full text or incomplete existing literature data;

(3) Low quality study. Newcastle Ottawa Quality Assessment Tool (NOS) score less than 5 points or AHRQ score less than 4 points indicates low quality.

## Search strategy

We will search the following databases: PubMed, Web of Science, CINAHL, Cochrane Library, EMBASE, CNKI, Wan-Fang, SinoMed and VIP. Furthermore, we will seek grey literature and mannually obtain the references cited in the article to ensure no relevant research is overlooked. This study will utilize medical subject headings (MeSH) and keywords for the search, encompassing the time from the foundation of the database to December 31, 2024. Comprehensive details regarding the search strategy are available in the attached file (S2 file). The search terms contain diabetes, pulmonary infection, pneumonia and influencing factors.

## Data collection and analysis

We will import all obtained studies into Endnote X9 software and eliminate all duplicate studies. Two trained researchers (W.H. and H.R.) will independently assess the titles and abstracts to eliminate studies that did not satisfy the inclusion criteria. Afterwards, two independent researchers (Z.Y. and J.Q.) will review the complete literature and exclude any that do not satisfy the criteria. If the researchers fail to reach consensus on the aforementioned two steps, the ultimate decision will be rendered by the third researcher (Y.R. or L.Y.). The research selection process is illustrated in Fig 1.

## Data extraction

Two researchers (W.H. and Z.Y.) will separately extract data utilizing pre-established tables. This table will gather the following information: Basic information (including author, country, and publication year); study characteristics (including sample size, age, gender, disease duration, laboratory test results, complications, and additional risk factors associated with pulmonary infection), and outcome effect data (such as incidence and timing of occurrence). If data is absent, we will reach out to the first author or corresponding author biweekly on Mondays and Fridays for a duration of two months. If no answer is obtained, the study will be incorporated into the research, but only a narrative summary will be presented.

## Assessment of risk of bias

Two qualified researchers (H.R. and J.Q.) will independently conduct a literature quality evaluation using the Newcastle Ottawa Scale (NOS) [19] to assess the literature quality of cohort studies and case-control studies. NOS includes 8 items, specifically including population selection, comparability, exposure/outcome evaluation. Except for comparability, which

can be rated up to 2 stars, all other items can be rated up to 1 star out of a total of 9 stars. The higher the score, the higher the research quality. The total score is 9 points. 0–4 indicates low quality, 5–6 indicates moderate quality, and 7–9 indicates high quality. Low quality studies with scores below 5 will be excluded. We will use The Agency for Healthcare Research and Quality (AHRQ) [20] to evaluate the literature quality of cross-sectional studies, which includes 11 items answered as' yes' (1 point), 'no' (0 points), or 'unclear' (0 points). The total score is 11 points. 0–3 indicates low quality, 4–7 indicates moderate quality, and 8–11 indicates high quality. Low quality studies with scores below 4 will be excluded. If there is any disagreement, the third researcher (Y.R. or L.Y.) will be consulted to make the final decision. The evaluation quality score of each study will be recorded in the basic information table of the study. This study will only include high-quality and medium quality literature, while the low-quality literature will be excluded. We will also use Grading of Recommendations, Assessment, Development and Evaluation (GRADE) to evaluate the accuracy of the meta-analysis results.

### Strategy for data synthesis

Our study will utilize RevMan 5.4 software to do a meta-analysis of risk factors identified in the literature collected. We will utilize Stata 16.0 to analysize data on influencing factors from three or more studies. The categorical variables are denoted by odds ratio (OR) and 95% confidence interval (CI), with $P < 0.05$ signifying statistically significant differences. The continuous data will be examined utilizing the standard mean deviation (SMD) or weighted mean deviation (WMD) with a 95% confidence interval (CI). In heterogeneity tests, $I^2 < 50\%$ and $P > 0.05$ indicate minimal heterogeneity, warranting the adoption of fixed effects models for analysis; conversely, $I^2 \geq 50\%$ and $P \leq 0.05$ signify the presence of heterogeneity. If heterogeneity remains $\geq 50\%$ after eliminating evident sources of clinical heterogeneity, a random effects model analysis is employed. Substantial heterogeneity will be examined utilizing Stata 16.0 software, focusing on age, gender, geography, sample size, and various risk factors via subgroup or sensitivity analysis. If heterogeneity exceeds 75%, We will conduct subgroup analyses (based on study design, population characteristics, etc.) to explore the sources of heterogeneity. If the heterogeneity still cannot be eliminated,a meta-analysis will not be performed. We shall employ descriptive analysis.

### Quality of evidence and publication biases assessment

Two researchers (WH and HR) will use the GRADE system to assess the quality of evidence, grading it based on five dimensions: risk of bias, inconsistency, indirectness, imprecision, and publication bias. The initial rating for randomized controlled trials is high quality, while observational studies are rated as low quality. The grades will be dynamically adjusted based on the assessment results of each dimension: when there is severe bias (such as methodological flaws), significant heterogeneity ($I^2 \geq 75\%$), deviation of PICO elements, confidence intervals crossing clinical thresholds, or publication bias, the grade will be downgraded; when a large effect size ($RR > 2 / < 0.5$) or dose-response relationship is found, the grade will be upgraded. For the assessment of publication bias, when the number of included studies is more than 10, the Egger test ($p < 0.05$ indicates the presence of bias) is used. If $p > 0.05$, the trim-and-fill method is employed to iteratively prune asymmetric extreme values, estimate missing studies, and recalculate the effect size to correct potential publication bias. The final evidence quality is classified into four levels: high, medium, low, and very low, corresponding to different recommendation intensities.

### Sensitivity analysis

In the heterogeneity test, when I2 is less than 50% and P is greater than 0.05, it indicates a relatively small degree of heterogeneity, and the fixed-effect model can be used for analysis; conversely, when I2 is greater than or equal to 50% and P is less than or equal to 0.05, it indicates the presence of heterogeneity. If after eliminating obvious clinical heterogeneity

factors, the heterogeneity is still greater than or equal to 50%, then the random-effect model should be used for analysis. For significant heterogeneity, we will use Stata 16.0 software for testing, focusing on age, gender, region, sample size, and various risk factors. Through subgroup analysis(According to the research design, population characteristics, etc.) or sensitivity analysis, the analysis will be conducted. If the heterogeneity exceeds 75%, meta-analysis will not be performed. We will use descriptive analysis. The funnel plot and Egger test ($\alpha = 0.1$) will be used to evaluate publication bias.

## Discussion

At present, many studies have analyzed the influencing factors of pulmonary infection in diabetes patients, but there are some differences in the results of each study. At present, there is no systematic evaluation to fully assess the risk factors of pulmonary infection in diabetes patients. Therefore, this study uses systematic review and meta-analysis to further understand the main factors that increase the risk of pulmonary infection in patients with type 2 diabetes.

There is an association between diabetes and decline of lung function [21]. It has been found that the worse the baseline pulmonary function, the higher the blood glucose level, the more severe the blood glucose fluctuation [22], and the faster the deterioration of pulmonary function in diabetes patients [6,23]. The reason may be that the blood glucose control of diabetes patients is unstable, and blood lipids are abnormal, leading to the impairment of carbon monoxide lung diffusion ability [24]. In addition, airflow restriction will affect blood sugar stability and further aggravate the condition of diabetes [25]. Therefore, Zhang et al proposed that more stringent blood glucose targets should be applied to diabetes patients to improve lung function [26].

The elderly with diabetes are at high risk of lung infection. With factors such as aging, respiratory muscle atrophy, and decreased lung elasticity, when pathogenic bacteria invade and infect the upper respiratory tract, the cough reflex is weak and it is difficult to cough up phlegm, leading to lung infection. And the lungs are also the habitat for various bacteria, such as Streptococcus pneumoniae, Haemophilus influenzae, Streptococcus pyogenes, etc. Long term hyperglycemia in the body environment of diabetes patients is also conducive to bacterial growth and reproduction. Therefore, it is necessary to conduct a meta-analysis on the risk factors of lung infection in diabetes patients, so as to provide suggestions for doctors to make clinical decisions.

## Supporting information

**S1 File. PRISMA-P.**
(DOCX)

**S2 File. Search strategy.**
(DOCX)

## Author contributions

**Conceptualization:** Wenhao Su.

**Data curation:** Luo Yang.

**Formal analysis:** Wenhao Su, Hairong Jia.

**Investigation:** Wenhao Su, Hairong Jia, Luo Yang, Jiaqi Zhang.

**Methodology:** Wenhao Su.

**Supervision:** Luo Yang, Yanru Wang.

**Writing – original draft:** Zhaoyang Wei, Wenhao Su, Hairong Jia, Yanru Wang.

**Writing – review & editing:** Wenhao Su, Luo Yang, Yanru Wang.

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
