## [Decision Letter · Decision Letter 0]

PONE-D-24-54675Risk factors for pulmonary infection in elderly patients with type 2 diabetes: a protocol for systematic review and meta-analysisPLOS ONE

Dear Dr. Yanru,

Thank you for submitting your manuscript to PLOS ONE. After careful consideration, we feel that it has merit but does not fully meet PLOS ONE’s publication criteria as it currently stands. Therefore, we invite you to submit a revised version of the manuscript that addresses the points raised during the review process.

We look forward to receiving your revised manuscript.

Kind regards,

Aida Fallahzadeh

Guest Editor

PLOS ONE

Journal Requirements:

Additional Editor Comments:

Dear Dr. Yanru,

Thank you for your submission of "Risk factors for pulmonary infection in elderly patients with type 2 diabetes: a protocol for systematic review and meta-analysis" to our journal. After careful consideration, I, along with the reviewer, believe that your manuscript has great potential, but there are several important revisions that need to be addressed before it can be accepted for publication.

The feedback from the reviewer is attached, and they have identified key areas for improvement. These revisions are substantial but necessary to enhance the overall quality and impact of your paper.

Kind regards

Aida Fallahzadeh, MD

Guest Editor

Reviewers' comments:

Reviewer's Responses to Questions

**Comments to the Author**

1. Does the manuscript provide a valid rationale for the proposed study, with clearly identified and justified research questions?

Reviewer #1: Partly

2. Is the protocol technically sound and planned in a manner that will lead to a meaningful outcome and allow testing the stated hypotheses?

Reviewer #1: Partly

3. Is the methodology feasible and described in sufficient detail to allow the work to be replicable?

Reviewer #1: Yes

4. Have the authors described where all data underlying the findings will be made available when the study is complete?

Reviewer #1: No

5. Is the manuscript presented in an intelligible fashion and written in standard English?

Reviewer #1: Yes

6. Review Comments to the Author

You may also provide optional suggestions and comments to authors that they might find helpful in planning their study.

Reviewer #1: This manuscript outlines a well-structured protocol for a systematic review and meta-analysis aimed at identifying risk factors for pulmonary infections in elderly patients with type 2 diabetes. The topic is highly relevant and addresses an important gap in the current literature. The authors have followed established guidelines such as PRISMA-P and registered their protocol in PROSPERO, enhancing the credibility and transparency of their study.

However, there are areas where the protocol could be improved to ensure methodological rigor and clarity. Below are my detailed comments.

Major concerns:

Definition of Pulmonary Infection:

The inclusion criteria for "diabetes pulmonary infection" are broad. Clarify how heterogeneity in diagnostic criteria (e.g., clinical diagnosis vs. self-report) will be managed.

Management of High Heterogeneity:

The protocol mentions descriptive analysis if heterogeneity exceeds 75%. Provide more details about how the authors plan to interpret and handle such cases.

Exclusion Criteria:

The exclusion of conference articles and protocols may omit valuable preliminary findings. Justify this decision or consider including these sources with appropriate caveats.

Minor Concerns

Subgroup Analysis:

While subgroup analyses are planned, specific subgroup categories (e.g., age groups, geographical regions) should be predefined to enhance clarity.

GRADE Framework:

Expand on how GRADE will be applied, particularly the domains assessed and the thresholds for grading evidence quality.

Publication Bias:

The "cut-and-patch approach" for publication bias requires further explanation to avoid ambiguity.

Pilot Search:

A pilot search to validate the feasibility and comprehensiveness of the search strategy would strengthen the protocol.

Figures and Appendices:

The PRISMA flow diagram (S1 Fig) should be included in the main document for better visibility and understanding.

Suggestions for Improvement:

Predefine risk factors and subgroup categories in the protocol.

Provide a rationale for excluding low-quality studies based on specific score thresholds.

Consider including conference articles or gray literature with appropriate quality checks.

Add a brief justification for the statistical tools (e.g., RevMan 5.4, STATA 16.0) used for analysis.

7. PLOS authors have the option to publish the peer review history of their article (what does this mean? ). If published, this will include your full peer review and any attached files.

**Do you want your identity to be public for this peer review?** For information about this choice, including consent withdrawal, please see our Privacy Policy .

Reviewer #1: No

---

## [Author Response · Author response to Decision Letter 1]

26 Jun 2025

Revised Manuscript with Track Changes

We are very grateful for your professional comments on our article. As you are concerned, there are several issues that need to be addressed. Based on your suggestions, we have made extensive corrections to the previous manuscript, and the specific corrections are as follows.

Reviewer #1

Definition of Pulmonary Infection:

This manuscript outlines a well-structured protocol for a systematic review and meta-analysis aimed at identifying risk factors for pulmonary infections in elderly patients with type 2 diabetes. The topic is highly relevant and addresses an important gap in the current literature. The authors have followed established guidelines such as PRISMA-P and registered their protocol in PROSPERO, enhancing the credibility and transparency of their study.

Response Thanks for the positive comments

Major concerns:

Definition of Pulmonary Infection:

The inclusion criteria for "diabetes pulmonary infection" are broad. Clarify how heterogeneity in diagnostic criteria (e.g., clinical diagnosis vs. self-report) will be managed.

Response Thank you for your valuable comments. We have carefully searched the relevant literature and further clarified the diagnostic criteria and definition of diabetes combined with pneumonia. At the same time, in order to standardize the heterogeneity of diagnostic criteria, we require that the patient's diagnosis must be based on some objective examination results. We have carefully revised the revised manuscript based on your comments.

This study was based on the Chinese Consensus for the Diagnosis and Treatment of Diabetes Complicated with Pneumonia. Patients with confirmed diabetes-related pulmonary infections through clinical diagnosis or self-reporting were included. To standardize the management of heterogeneity in diagnostic criteria, we required that the diagnostic basis of the patients must include: 1. Fasting blood glucose ≥ 7.0 mmol/L or random blood glucose ≥ 11.1 mmol/L and confirmed pulmonary infection by CT; 2. The clinical case needs to include CURB-65 score, inflammatory markers, and qualified pathogen evidence.

Please refer to the "Participants" section.

Management of High Heterogeneity:

The protocol mentions descriptive analysis if heterogeneity exceeds 75%. Provide more details about how the authors plan to interpret and handle such cases.

Response We think this is a good suggestion. We have improved the solution for handling high heterogeneity in the revised manuscript. If the heterogeneity exceeds 75%, we will explore the source of heterogeneity through subgroup analysis (according to study design, population characteristics, etc.). If the heterogeneity still cannot be eliminated, no meta-analysis will be performed. We will use descriptive analysis.

If heterogeneity exceeds 75%,We will conduct subgroup analyses (based on study design, population characteristics, etc.) to explore the sources of heterogeneity. If the heterogeneity still cannot be eliminated,a meta-analysis will not be performed. We shall employ descriptive analysis.

Please refer to the "Strategy for Data Synthesis" section.

Exclusion Criteria:

The exclusion of conference articles and protocols may omit valuable preliminary findings. Justify this decision or consider including these sources with appropriate caveats.

Response We thank the reviewers for their valuable comments. This study chose to exclude conference abstracts and research plans based on the following considerations: Methodological rigor requirements:

1.Conference abstracts often lack complete methodological details and original data, making it difficult to conduct quality assessments (such as NOS/AHRQ scores), which may introduce bias risks. The research protocol is prospectively designed and has not yet produced analyzable research results.

2.Matching of evidence levels: According to the GRADE system, conference literature is classified as "low-level evidence", which is different from the "at least moderate-quality evidence" required by the research objectives. Retaining such literature may reduce the overall strength of evidence.

Minor Concerns

Subgroup Analysis:

While subgroup analyses are planned, specific subgroup categories (e.g., age groups, geographical regions) should be predefined to enhance clarity.

Response We appreciate your reminder and we have added the specific subgroup analysis categories in the revised manuscript.

If heterogeneity exceeds 75%,We will conduct subgroup analyses (based on study design, population characteristics, etc.) to explore the sources of heterogeneity. If the heterogeneity still cannot be eliminated,a meta-analysis will not be performed. We shall employ descriptive analysis.

Through subgroup analysis(According to the research design, population characteristics, etc.) or sensitivity analysis, the analysis will be conducted. If the heterogeneity exceeds 75%, meta-analysis will not be performed.

Please refer to the "Sensitivity Analysis" and "Strategy for Data Synthesis"

GRADE Framework:

Expand on how GRADE will be applied, particularly the domains assessed and the thresholds for grading evidence quality.

Response:We thank the reviewer for drawing our attention to this issue. We have systematically supplemented the application of the GRADE method in the revised manuscript, which mainly includes the following key contents: 1. The grading of the GRADE system assessment quality; 2. The GRADE scoring criteria; 3. A detailed description of how to use the GRADE system to assess the quality of evidence.

Two researchers (WH and HR) will use the GRADE system to assess the quality of evidence, grading it based on five dimensions: risk of bias, inconsistency, indirectness, imprecision, and publication bias. The initial rating for randomized controlled trials is high quality, while observational studies are rated as low quality. The grades will be dynamically adjusted based on the assessment results of each dimension: when there is severe bias (such as methodological flaws), significant heterogeneity (I² ≥ 75%), deviation of PICO elements, confidence intervals crossing clinical thresholds, or publication bias, the grade will be downgraded; when a large effect size (RR > 2/< 0.5) or dose-response relationship is found, the grade will be upgraded. For the assessment of publication bias, when the number of included studies is more than 10, the Egger test (p < 0.05 indicates the presence of bias) is used. If p > 0.05, the trim-and-fill method is employed to iteratively prune asymmetric extreme values, estimate missing studies, and recalculate the effect size to correct potential publication bias. The final evidence quality is classified into four levels: high, medium, low, and very low, corresponding to different recommendation intensities.

In the heterogeneity test, when I2 is less than 50% and P is greater than 0.05, it indicates a relatively small degree of heterogeneity, and the fixed-effect model can be used for analysis; conversely, when I2 is greater than or equal to 50% and P is less than or equal to 0.05, it indicates the presence of heterogeneity. If after eliminating obvious clinical heterogeneity factors, the heterogeneity is still greater than or equal to 50%, then the random-effect model should be used for analysis. For significant heterogeneity, we will use Stata 16.0 software for testing, focusing on age, gender, region, sample size, and various risk factors. Through subgroup analysis(According to the research design, population characteristics, etc.) or sensitivity analysis, the analysis will be conducted. If the heterogeneity exceeds 75%, meta-analysis will not be performed. We will use descriptive analysis. The funnel plot and Egger test (α = 0.1) will be used to evaluate publication bias.

Please refer to the "Quality of evidence and publication biases assessment" and "Sensitivity analysis"

Publication Bias:

The "cut-and-patch approach" for publication bias requires further explanation to avoid ambiguity.

Response: We thank the reviewer for the suggestion. We have added the “deletion and repair method” explanation of publication bias in the revised manuscript to avoid ambiguity.

Two researchers (WH and HR) will use the GRADE system to assess the quality of evidence, grading it based on five dimensions: risk of bias, inconsistency, indirectness, imprecision, and publication bias. The initial rating for randomized controlled trials is high quality, while observational studies are rated as low quality. The grades will be dynamically adjusted based on the assessment results of each dimension: when there is severe bias (such as methodological flaws), significant heterogeneity (I² ≥ 75%), deviation of PICO elements, confidence intervals crossing clinical thresholds, or publication bias, the grade will be downgraded; when a large effect size (RR > 2/< 0.5) or dose-response relationship is found, the grade will be upgraded. For the assessment of publication bias, when the number of included studies is more than 10, the Egger test (p < 0.05 indicates the presence of bias) is used. If p > 0.05, the trim-and-fill method is employed to iteratively prune asymmetric extreme values, estimate missing studies, and recalculate the effect size to correct potential publication bias. The final evidence quality is classified into four levels: high, medium, low, and very low, corresponding to different recommendation intensities.

Please refer to the Quality of evidence and publication biases assessment

Pilot Search:

A pilot search to validate the feasibility and comprehensiveness of the search strategy would strengthen the protocol.

Response: Thank you for your careful review of the research methods. Regarding the suggestions for preliminary search, we have carefully considered that as a research protocol, the current version already contains a pre-tested search strategy (Supplementary File), and considering that the core purpose of the protocol is to preset the method framework rather than to verify, its effectiveness will be fully presented through the PRISMA flow chart of the formal systematic review stage.

Figures and Appendices:

The PRISMA flow diagram (S1 Fig) should be included in the main document for better visibility and understanding.

Response: Thank you for the reviewer's detailed comments. We have included the PRISMA flow diagram (S1 Fig) in the main text of the revised manuscript.

Cover Letter

Dear editor

On behalf of all the authors involved in the writing of the article titled "Risk Factors for Pulmonary Infections in Elderly Patients with Type 2 Diabetes: Systematic Review and Meta-analysis Protocol" (PONE-D-24-54675), I would like to express my sincere gratitude to you and the reviewers for your valuable comments and constructive suggestions. I have carefully considered these comments and have revised the paper based on their opinions. These comments are extremely valuable and helpful for us to improve this article. We have made numerous revisions to the manuscript according to the editors' and reviewers' opinions. We hope these revisions will be satisfactory. During this revision process, based on the actual contributions of each author, after all-party consensus, we adjusted the author order. Wei Chaoyang's position was moved forward because he made significant contributions to this revised paper. Once again, we thank you for considering publishing our paper in your journal. We look forward to receiving your reply as soon as possible.

Best wishes

Sincerely,

Wang Yanru

Zhejiang Chinese Medical University

Wangyanru001@outlook.com

June 24th, 2025

---

## [Editor Report · Decision Letter 1]

Risk factors for pulmonary infection in elderly patients with type 2 diabetes: a protocol for systematic review and meta-analysis

PONE-D-24-54675R1

Dear Dr. Wang,

We’re pleased to inform you that your manuscript has been judged scientifically suitable for publication and will be formally accepted for publication once it meets all outstanding technical requirements.

Kind regards,

Aida Fallahzadeh

Guest Editor

PLOS ONE

Additional Editor Comments (optional):

Dear Authors,

Thank you for thoroughly addressing all the comments raised during the review process. I found your responses clear and satisfactory. I am pleased to inform you that the manuscript has been accepted for publication.

Bests,

Aida Fallahzadeh, MD

Guest Editor
---

## [Editor Report · Acceptance letter]

PONE-D-24-54675R1

PLOS ONE

Dear Dr. Wang,

I'm pleased to inform you that your manuscript has been deemed suitable for publication in PLOS ONE. Congratulations! Your manuscript is now being handed over to our production team.

Kind regards,

on behalf of

Dr. Aida Fallahzadeh

Guest Editor

PLOS ONE